# Enhanced Levels of Peroxisome-Derived H_2_O_2_ Do Not Induce Pexophagy but Impair Autophagic Flux in HEK-293 and HeLa Cells

**DOI:** 10.3390/antiox12030613

**Published:** 2023-03-02

**Authors:** Hongli Li, Celien Lismont, Cláudio F. Costa, Mohamed A. F. Hussein, Myriam Baes, Marc Fransen

**Affiliations:** 1Laboratory of Peroxisome Biology and Intracellular Communication, Department of Cellular and Molecular Medicine, Katholieke Universiteit Leuven, 3000 Leuven, Belgium; 2Department of Biochemistry, Faculty of Pharmacy, Assiut University, Asyut 71515, Egypt; 3Laboratory of Cell Metabolism, Department of Pharmaceutical and Pharmacological Sciences, Katholieke Universiteit Leuven, 3000 Leuven, Belgium

**Keywords:** peroxisomes, hydrogen peroxide, pexophagy, autophagy, oxidative stress, mKeima, optineurin

## Abstract

Peroxisomes are functionally specialized organelles that harbor multiple hydrogen peroxide (H_2_O_2_)-producing and -degrading enzymes. Given that this oxidant functions as a major redox signaling agent, peroxisomes have the intrinsic ability to mediate and modulate H_2_O_2_-driven processes, including autophagy. However, it remains unclear whether changes in peroxisomal H_2_O_2_ (po-H_2_O_2_) emission impact the autophagic process and to which extent peroxisomes with a disturbed H_2_O_2_ metabolism are selectively eliminated through a process called “pexophagy”. To address these issues, we generated and validated HEK-293 and HeLa pexophagy reporter cell lines in which the production of po-H_2_O_2_ can be modulated. We demonstrate that (i) po-H_2_O_2_ can oxidatively modify multiple selective autophagy receptors and core autophagy proteins, (ii) neither modest nor robust levels of po-H_2_O_2_ emission act as a prime determinant of pexophagy, and (iii) high levels of po-H_2_O_2_ impair autophagic flux by oxidative inhibition of enzymes involved in LC3II formation. Unexpectedly, our analyses also revealed that the autophagy receptor optineurin can be recruited to peroxisomes, thereby triggering pexophagy. In summary, these findings lend support to the idea that, during cellular and organismal aging, peroxisomes with enhanced H_2_O_2_ release can escape pexophagy and downregulate autophagic activity, thereby perpetuating the accumulation of damaged and toxic cellular debris.

## 1. Introduction

Autophagy is an evolutionarily conserved multistep process that delivers superfluous or damaged components of the cytoplasm, including organelles, to the lysosome for turnover [1]. During macroautophagy, the major type of autophagy in mammals, the cargo is engulfed by a double-membrane vesicle, called the autophagosome [2]. From a mechanistic point of view, this process—which is orchestrated by an arsenal of lipids, lipid-metabolizing enzymes, and autophagy-related (ATG) proteins—can be divided into multiple stages: initiation by cell signaling pathways; phagophore nucleation, expansion, and closure; and autophagosome fusion with the lysosome [3]. Autophagy can occur in either a non-selective or selective manner, with the latter requiring the additional action of specific autophagy receptors (SARs) (e.g., SQSTM1, NBR1, OPTN, and CACO2) [4]. In general, these receptors recognize molecular tags (e.g., a conjugated ubiquitin) on their cargo and direct them to nascent autophagic membranes by direct interaction with members of the ATG8 family, such as LC3B [5].

Autophagy is a redox-regulated process that can be activated by oxidative stress, thereby protecting cells from the accumulation of oxidatively damaged macromolecules and organelles [6,7,8]. This regulation is governed at the transcriptional and posttranslational levels [9,10,11]. A key signaling messenger in autophagy regulation is hydrogen peroxide (H_2_O_2_), which has the capacity to promote disulfide bond formation within or between proteins (or between proteins and small molecules such as glutathione and coenzyme A) through the oxidation of redox-sensitive cysteine residues [12,13]. This, in turn, can induce conformational changes that may affect the stability, subcellular localization, and/or activity of the affected proteins. Examples within the context of autophagy include proteins involved in (i) gene transcription (e.g., FOXO [14] and TFEB [15]), (ii) autophagy initiation and phagophore nucleation (e.g., PTEN [16], ATM [17], and TSC2 [11]), (iii) phagophore expansion (e.g., ATG3 [18], ATG4A [19], ATG4B [20], and ATG7 [18]), (iv) cargo recognition (e.g., OPTN [21,22], SQSTM1 [23,24], and CACO2 [22]), and (v) autophagosome maturation (e.g., cathepsin L [25]). Importantly, whether disulfide bond formation enhances or impedes the activity of these proteins is protein- and context-specific. Examples relevant to this study are briefly discussed in the next paragraph. For a more in-depth discussion of these and other examples, we refer the reader to a recent review [6].

ATG4 proteins exert a dual role in autophagy: on the one hand, they activate LC3 by cleavage of pro-LC3, thereby exposing a C-terminal glycine residue that is required for conjugation to phosphatidyl-ethanolamine (PE) at the autophagosome membrane; on the other hand, they are necessary for degreasing LC3-PE (termed LC3-II) to a cytosolic isoform (termed LC3-I). Interestingly, disulfide bond formation of the human orthologs of ATG4 has been shown to render these cysteine proteases enzymatically inactive [19,20]. This implies that, under conditions of oxidative stress, their activities need to be tightly controlled to ensure autophagy progression [26]. In addition, under basal conditions, the E1-like and E2-like enzymes ATG7 and ATG3 form inactive thioester-bonded complexes with LC3-I. On stimulation of autophagy, these ATGs dissociate from LC3-I, thereby freeing their catalytic thiols. However, under severe oxidative stress conditions, the non-LC3-shielded thiols in ATG3 and ATG7 can form intermolecular disulfide bridges, thereby preventing LC3 lipidation, autophagosome maturation, and autophagy [18]. Finally, disulfide-bond-mediated multimerization of SQSTM1 and CACO2 has been shown to facilitate the recruitment of the autophagy machinery to cargo (e.g., mitochondria) through high-avidity binding to LC3-II on nascent autophagic membranes [23], thereby inducing pro-survival autophagy (or mitophagy) under oxidative stress conditions [22,24].

Peroxisomes are dynamic organelles that rapidly adapt their protein content, morphology, and number in response to cellular needs and environmental cues [27]. In mammals, these organelles are best known for their role in cellular lipid metabolism [28]. However, given that peroxisomes are equipped with a panel of flavin-containing oxidases, which reduce molecular oxygen to H_2_O_2_ as part of their catalytic cycle, and catalase, an enzyme that decomposes H_2_O_2_, they have the intrinsic ability to initiate and modulate H_2_O_2_-driven signaling events, including (selective) autophagy [29]. To maintain peroxisomal fitness, it is crucial that damaged as well as surplus organelles are selectively removed. Currently, it is commonly accepted that in mammals, this process, called pexophagy, primarily occurs through the autophagy-lysosome pathway. Pexophagy can occur through both ubiquitin-dependent and -independent mechanisms, and known triggers include amino acid depletion, hypoxia, viral and bacterial infections, dysfunctional peroxisome biogenesis, and oxidative stress [30,31]. However, current knowledge about the signaling pathways behind pexophagy is sparse and only recently emerging, in particular, in mammalian cells [32].

The aim of this study was to investigate how the production of peroxisome-derived H_2_O_2_ (po-H_2_O_2_) affects general autophagy and pexophagy. Here, it is pertinent to mention that we recently developed a unique and powerful Flp-In T-REx 293 cell model that allows the selective generation of H_2_O_2_ inside peroxisomes in a time- and dose-dependent manner (i) through controlled expression of a destabilization domain (DD)-tagged version of D-amino acid oxidase (DAO), and (ii) by regulating its activity with the supplementation of a D-amino acid (e.g., D-Ala) to the assay medium [33]. In this study, we used this DD-DAO Flp-In T-REx 293 cell line as well as a similar HeLa cell model as a starting point to generate cell lines stably expressing mRFP-GFP-LC3, an autophagic flux indicator [34], or po-mKeima, a peroxisome-targeted variant of the pH-sensitive red fluorescent protein mKeima [35], and to subsequently investigate how controlled release of po-H_2_O_2_ impacted pexophagy and general autophagy.

## 2. Materials and Methods

### 2.1. DNA Manipulations and Plasmids

The plasmid encoding po-mKeima (pMF2005) was generated by amplifying the corresponding cDNA via PCR (template: mKeima-Red-N1 (Addgene plasmid 54597; gift from Dr. M. Davidson (Florida State University, Tallahassee, FL, USA)); primers: 5′-gggagatctaccatggtgagcgtgatcgccaagcag-3’ and 5′-tccccgcggccgc**ttacagcttgctctt**gcccagcagggagtggcgggcgatg-3’ (the restriction enzyme recognition sites and nucleotides encoding the prototypic C-terminal PTS1 “-KSKL” are underlined and indicated in bold, respectively)) and ligating the Bgl II/Not I-digested PCR product into Bgl II/Not I-restricted pEGFP-N1 (Clontech, Mountain View, CA, USA; 6085-1). The plasmid encoding OPTN-GFP was obtained from Addgene (plasmid 27052; gift from Dr. B. Yue (University of Illinois, Chicago, IL, USA) [36]). The plasmids encoding po-roGFP2 (pMF1706) [37], c-roGFP2-Orp1 (pMF1834) [33], mCherry-PTS1 (pMF1218) [38], or GFP_Nb_-PM (pMF1968) [39] have been described elsewhere. The plasmid encoding mRFP-GFP-LC3 (p4-RFP-GFP-LC3) was kindly provided by Dr. G. Bultynck (KU Leuven, Leuven, Belgium). The *TOP10F’ E. coli* strain (Thermo Fisher Scientific, San Jose, CA, USA; C3030-06) was used as a cloning and plasmid amplification host. All plasmids were validated by DNA sequencing (LGC Genomics, Berlin, Germany).

### 2.2. Cell Culture and Treatment

The DD-DAO Flp-In T-REx 293 and HeLa cell lines were generated as detailed elsewhere [33]. Cells were cultured at 37 °C in a humidified 5% CO_2_ incubator in regular minimum essential medium Eagle α (rMEMα; Alpha MEM (Lonza, BE12-169F), 10% (*v*/*v*) fetal bovine serum (FBS; Biowest, S181B), 2 mM UltraGlutamine I (Lonza, BE17-605E/U1), and 0.2% (*v*/*v*) MycoZap (Lonza, Verviers, Belgium; VZA-2012)). Electroporations were performed with the Neon Transfection System (Thermo Fisher Scientific) and conducted in 10 µL tips with the following electroporation parameters: 1150 V, 20 ms pulse width, 2 pulses (for 293 cells); 1500 V, 20 ms pulse width, 1 pulse (for HeLa cells; kindly provided by Dr. M. Bollen (KU Leuven, Belgium)). To generate DD-DAO Flp-In T-REx cell lines constitutively expressing po-mKeima or mRFP-GFP-LC3, the cells were transfected with pMF2005 or p4-RFP-GFP-LC3, respectively, and cultured for 2–3 weeks in rMEMα supplemented with 10 µg/mL blasticidin (InvivoGen, San Diego, CA, USA; ant-bl-1), 100 µg/mL hygromycin (InvivoGen, ant-hg-1), and 200 µg/mL (for po-mKeima) or 400 µg/mL (for mRFP-GFP-LC3) G418 (Thermo Fisher Scientific, BP673-5). Stably transformed clones of flow-sorted pools (Sony, Surrey, UK; MA900) were selected and further cultured in the same medium containing 200 μg/mL of G418. 

To selectively express DD-DAO in peroxisomes, DD-DAO Flp-In T-REx cells were cultured for 3 days in rMEMα containing 1 μg/mL doxycycline (DOX; Sigma, St. Louis, MO, USA; D9891) and 500 nM Shield1 (Clontech, 632189) and subsequently for 1 day in the same medium lacking DOX/Shield1 [33]. 

To generate H_2_O_2_ inside peroxisomes, the DOX/Shield1-chased cells were incubated in (i) DPBS (Lonza, BE17-512F) supplemented with 10 mM 3-amino-1,2,4-triazole (3-AT; Thermo Fisher Scientific, 264571000), (ii) modified DMEM (mDMEM) lacking glucose and glutamine (Gibco, Dublin, Ireland; A144300) but supplemented with 5% FBS, 2 mM UltraGlutamine I, 10 mM sodium pyruvate (Gibco, 11360070), 10 mM 3-AT, and 50 µM dehydroepiandrosterone (DHEA; Sigma, 700087P), or (iii) modified MEMα (mMEMα) containing 10 mM 3-AT and 200 µM DHEA, always in combination with 20 mM L- or D-alanine (L/D-Ala) as indicated. 

For starvation, the cells were first washed three times with Earle’s balanced salt solution (EBSS; Sigma, E2888) and subsequently incubated in the same buffer for the indicated time. To stimulate autophagy, the cells were treated for 8 or 14 h with 1 µM Torin-1 (Bio-techne, Milan, Italy; 4247) or 50 nM rapamycin (MCE, South Brunswick, NJ, USA; AY 22989), respectively; to block autophagosome formation, the cells were pretreated for 24 h with 5 mM 3-methyladenine (3-MA; Selleckchem, Houston, TX, USA; S2767), and to prevent autophagosome-lysosome fusion, the cells were pretreated for 2 or 8 h with 200 µM chloroquine (CQ; Sigma, C6628) or 200 nM bafilomycin A1 (BafA1; Cayman Chemical, Ann Arbor, MI, USA; 11038), respectively. The same volume of DMSO (MP Biomedicals, Solon, OH, USA; 02196055) was included as the vehicle control. To image lysosomes, the cells were incubated with LysoTracker Green DND-26 (Invitrogen, Merelbeke, Belgium; L7526). 

To investigate the effect of acidic pH on po-mKeima, the cells were incubated in a citrate buffer (135 mM KCl, 2 mM K_2_HPO_4_, 9.1 mM sodium citrate (dihydrate), 10.9 mM citric acid, 1.2 mM CaCl_2_, 0.8 mM MgSO_4_; pH 4.5).

### 2.3. Antibodies

The following primary antibodies were used for immunofluorescence (IF) or immunoblotting (IB): rabbit anti-ATG3 (Abcam, Cambridge, UK; ab108251; IF, 1:50; IB, 1:1000), anti-ATG4B (CST, Leiden, The Netherlands; 5299; IF, 1:100; IB, 1:1000), anti-ATG7 (Abcam, ab133528; IB, 1:1000), anti-SQSTM1 (Abcam, ab109012; IF, 1:200; IB, 1:20,000), anti-OPTN (Abcam, ab23666; IF, 1:50; IB, 1:1000), anti-CACO2/NDP52 (Abcam, ab68588; IF, 1:100; IB, 1:2000), anti-LC3B (CST, 2775; IB, 1:1000), anti-TUBA1A (Santa Cruz Biotechnology, Heidelberg, Germany; SC-5546; IB, 1:1000), anti-USP30 (Abbkine Scientific, Wuhan, China; ABP52679; IB, 1:1000), anti-DNM1L (Cusabio, Houston, TX, USA; CSB-PA002203; IB, 1:1000), mouse anti-PEX14 (IF, 1:100) [40], and anti-actin (Sigma, A5316; IB, 1:10,000). The secondary antibodies for IF were conjugated to Alexa Fluor 488 (Invitrogen, A11017; 1:2000) or Texas Red (Calbiochem, Darmstadt, Germany; 401355; 1:200), and the secondary antibodies for IB were conjugated to alkaline phosphatase (Sigma, A3687 and A2429; 1:5000 and 1:10,000, respectively).

### 2.4. Immunoblotting

Sample preparation for IB was carried out as previously described [33]. A modified protocol was used for cell lysates. Specifically, the lysates were collected in RIPA lysis buffer (Sigma, R0278) supplemented with a protease inhibitor mix (Sigma, P2714). We added 5X sample buffer and the samples were boiled at 100 °C for 10 min. A freshly prepared 5-bromo-4-chloro-3-indolyl-phosphate (GERBU, Heidelberg, Germany; #1735)/nitro blue tetrazolium solution (GERBU, #1015) in alkaline phosphatase buffer (100 mM Tris-HCl, pH 9.2; 100 mM NaCl; 5 mM MgCl_2_) was used as the staining substrate.

### 2.5. Fluorescence Microscopy

Fluorescence microscopy was essentially carried out as described previously [41]. The following filter cubes were chosen to match the properties of the fluorescent reporters used in this study: F400 (excitation: 390–410 nm; dichroic mirror: 505 nm; emission: 510–550 nm), F440 (excitation: 422–432 nm; dichroic mirror: 600 nm; emission: 610 long pass), F482 (excitation: 470–495 nm; dichroic mirror: 505 nm; emission: 510–550 nm), and F562 (excitation: 545–580 nm; dichroic mirror: 600 nm; emission: 610 nm long pass). For live-cell imaging, cells were seeded and imaged in FluoroDish cell culture dishes (World Precision Instruments, Hertfordshire, UK; FD-35). To enhance the adherence of the Flp-In T-REx 293 cells to the glass surface, the dishes were precoated with polyethyleneimine (PEI) (MP Biomedicals, 195444) at 25 μg/mL in 150 mM NaCl (2 h, room temperature) [33]. For immunofluorescence microscopy, the samples were fixed and processed as described elsewhere [42]. Nanobody-based plasma membrane translocation assays were carried out as described elsewhere [39]. cellSens Dimension software (version 2.1) (Olympus Belgium) was used for image acquisition and analysis.

### 2.6. Electrophoretic Mobility Shift Assay

Electrophoretic mobility shift assays (EMSAs) were essentially performed as described previously [33]. A slightly modified protocol was used for cell lysate preparation. Specifically, the cell pellets were dissolved in RIPA lysis buffer containing 10 mM NEM. For the non-reducing samples, 5X SDS-PAGE sample buffer without a reducing agent was added and the samples were heated to 65 °C for 10 min; for the reducing samples, 5X SDS-PAGE containing 10% (*v*/*v*) 2-mercaptoethanol (Sigma, M7522) was added and samples were heated to 100 °C for 10 min.

### 2.7. Flow Cytometry Analysis

The cells were washed with DPBS and dislodged from the plate by trypsinization. Next, the cells were pelleted (300× *g*, 1 min), resuspended in the appropriate assay buffer (as specified elsewhere), and analyzed using flow cytometry (BD Biosciences, San Jose, CA, USA; FACSymphony™ A1 or A5). For po-mKeima, the cells were analyzed by using 405- and 561-nm lasers in combination with a 600–620 nm emission filter; for mRFP-GFP-LC3, the cells were analyzed by using 488 and 561 nm lasers in combination with 515–545 nm and 600–620 nm emission filters, respectively. For each sample, 100,000 events were collected and analyzed using BD FACSDiva v8.0.1 software (BD Biosciences).

### 2.8. Statistical Analysis

Statistical analysis was performed by one-way analysis variance (ANOVA) followed by Tukey’s post hoc test for multiple comparisons, and Student’s unpaired two-tailed *t*-test was used to compare two groups. All statistical analyses were conducted using GraphPad Prism (version 9.0.0 for Windows 64-bit, GraphPad Software, San Diego, CA, USA). Data are expressed as the mean ± standard deviation (SD). A *p*-value less than 0.05 was considered statistically significant.

## 3. Results

### 3.1. Validation of the DD-DAO/po-mKeima Flp-In T-REx Cell Lines

To establish robust and sensitive cell models that can be used to study the effects of po-H_2_O_2_ on pexophagy, we generated DD-DAO Flp-In T-REx 293 and HeLa cell lines stably expressing a peroxisome-targeted variant of mKeima (po-mKeima). The excitation spectrum of this coral-derived monomeric red fluorescent protein (emission peak: 620 nm) is pH-dependent (peak at neutral pH: 440 nm; peak at acidic pH: 586 nm) [43], a property that can be exploited to monitor the delivery of subcellular structures to autolysosomes [44]. A schematic of mKeima-based detection of pexophagy in living cells is depicted in Figure 1A. Importantly, colocalization studies with established markers clearly indicate that, on dual excitation of po-mKeima at high (around 586 nm; false color: red) versus low (around 440 nm; false color: green) wavelengths, the yellowish and reddish dots represent peroxisomes and autolysosomes, respectively (Figure 1B, Appendix A). Note also the substantial heterogeneity in the number of autolysosome-sequestered peroxisomes between individual cells (Figure 1C).

To functionally validate po-mKeima, DD-DAO/po-mKeima Flp-In T-REx 293 cells were subjected to starvation, a stressor known to induce pexophagy [45]. To distinguish between true pexophagy and overall acidification of the peroxisome lumen, we also included a condition in which the cells were incubated for 1 h in a citrate buffer (pH 4.5). While culturing the cells for 24 h in EBSS was associated with a varied but marked increase in acidic puncta, a phenomenon that could largely be counteracted by the autophagy inhibitor 3-methyladenine, exposing the cells to an acidic buffer resulted in an increase in fluorescence throughout the acidic channel (Figure 2A). To objectively quantify these changes, the cells were also analyzed by flow cytometry. From these data, it is clear that (i) an increase in pexophagy is accompanied by a decrease in fluorescence intensity in the neutral channel (Figure 2B), and (ii) global acidification of the peroxisome lumen leads to an increase in fluorescence intensity in the acidic channel (Figure 2C). Together, these observations corroborate the notion that po-mKeima is a suitable pexophagy reporter.

### 3.2. Peroxisomal H_2_O_2_ Emission Does Not Act as the Primary Trigger for Pexophagy in HEK-293 and HeLa Cells

To investigate to what extent po-H_2_O_2_ emission impinges on pexophagy, we first established assay conditions yielding different net levels of po-H_2_O_2_ in the cytosol. To that end, the DOX/Shield1-chased DD-DAO/po-mKeima Flp-In T-REx cells were incubated in different assay media: (i) DPBS supplemented with 10 mM 3-amino-1,2,4-triazole (3-AT; this is an inhibitor of catalase activity), indicated with “DPBS”, and 10 mM L- or D-Ala; (ii) modified DMEM (mDMEM) lacking glucose and glutamine but supplemented with 10 mM 3-AT, 50 μM dehydroepiandrosterone (DHEA; this is an inhibitor of glucose-6-phosphate dehydrogenase, the rate-limiting enzyme in the pentose phosphate pathway), and 20 mM L- or D-Ala; or (iii) modified MEMα (mMEMα) containing 10 mM 3-AT, 200 μM DHEA, and 20 mM L- or D-Ala. Note that the L-Ala conditions served as a negative control. To document the release of po-H_2_O_2_ on the addition of D-Ala, we used a cytosolic (c-) variant of the ratiometric H_2_O_2_ biosensor roGFP2-Orp1 [46]. On oxidation, the excitation maximum of this sensor shifts from 480 nm to 400 nm, and, by calculating the corresponding emission intensities at 516 nm, variations in the H_2_O_2_ levels can be assessed.

As shown in Figure 3, Appendix A, different assay media exhibited significant differences in cytosolic antioxidant capacity. Unexpectedly, we could not detect changes in pexophagy in any of the conditions studied, neither in HEK-293 (Figure 4A,B) nor in HeLa (Appendix A) cells. However, quantification of the results using flow cytometry revealed that the production of H_2_O_2_ inside peroxisomes slightly acidified the organelle lumen (Figure 4C,D), a phenomenon that went unnoticed with the microscopy-based approach (Figure 4A,B). In summary, these findings demonstrate that po-H_2_O_2_ does not act as a primary trigger for pexophagy, at least not in the cell lines and conditions evaluated.

Finally, we also cultured DD-DAO/po-mKeima Flp-In T-REx 293 cells in rMEMα supplemented or not with 10 mM 3-AT. However, under those conditions, no pexophagy induction could be observed (Appendix A).

### 3.3. Enhanced Levels of Peroxisome-Derived H_2_O_2_ Cause Late-Stage Impairment in Autophagy

To investigate whether or not an increase in po-H_2_O_2_ emission affects general autophagy, we used DD-DAO Flp-In T-REx 293 cells transiently or stably expressing mRFP-GFP-LC3. To validate this autophagic flux probe in our model system, we starved the cells for 8 h in EBSS containing or not containing the late-stage autophagy inhibitor BafA1. As expected [34], this treatment was followed by an increase in the number of autolysosomes (red puncta) or autophagosomes (yellow puncta) in the absence or presence of BafA1, respectively (Appendix A). In line with this, quantification by flow cytometry showed that, with starvation for 8 h, the proportion of cells with reduced green fluorescence intensity was increased by approximately 20%, but only in the condition without BafA1 (Appendix A).

Unexpectedly, neither mild (Figure 5A) nor moderate (Figure 5B,C) or acute (Figure 5D) increases in po-H_2_O_2_ emission enhanced general autophagic flux, as revealed by FACS analysis (representative fluorescence overlay images of mRFP-GFP-LC3 are shown in Appendix A). This was further evidenced by the observation that moderate levels of po-H_2_O_2_, in contrast to treatment with the mTOR inhibitor Torin-1, had no effect on the processing of endogenous LC3B-I to LC3B-II (Figure 6 and Appendix A). Interestingly, excessive emission of po-H_2_O_2_ was accompanied by an increase in the LC3B-II/I ratio (Figure 7A and Appendix A). However, given that such treatment did not increase the LC3B-II levels in cells precultured in the presence of BafA1 (Figure 7B and Appendix A), it seems elevated amounts of po-H_2_O_2_ do not promote but rather inhibit autophagy at a later stage.

### 3.4. The Autophagy Receptors SQSTM1, CACO2, and OPTN Aggregate in Response to Peroxisome-Derived H_2_O_2_

Given that it has been demonstrated that the autophagy receptors SQSTM1 [24], CACO2 [22], and OPTN [21,22] are redox-sensitive, we also checked whether these SARS are targets of po-H_2_O_2_. By employing immunofluorescence, we could demonstrate that po-H_2_O_2_ can promote receptor clustering (Figure 8), but not on peroxisomes (Appendix A). However, under basal conditions, we sporadically observed colocalization between OPTN and peroxisomes (see Section 3.6). Electrophoretic mobility shift (EMSA) analyses in the absence or presence of a reducing agent also showed that the higher-molecular-weight species formed in response to po-H_2_O_2_ are disulfide-bonded (Figure 9). Note that several other proteins tested under the same conditions (e.g., DNM1L, TUBA1A, and USP30) did not form such higher-molecular-weight complexes (Appendix A).

### 3.5. The Core Autophagy Machinery Proteins ATG3, ATG4B, and ATG7 Are Prone to Oxidation by Peroxisome-Derived H_2_O_2_

Next, we carried out redox EMSAs to assess whether po-H_2_O_2_ can oxidatively modify ATG3, ATG4B, and ATG7, whose conformations and activities have been shown to be redox-regulated [18,20]. The results provide direct evidence that these proteins are also targets of po-H_2_O_2_ (Figure 10), thereby confirming and extending prior observations that (i) under basal conditions, ATG3 predominantly exists as a disulfide-bonded ±50 kDa complex, corresponding to LC3-bound ATG3 [18], which is converted into multiple higher-molecular-weight complexes under conditions of oxidative stress (Figure 10A), (ii) under basal conditions, ATG4 is mainly present in an intramolecular disulfide-stabilized conformation that can be rearranged with oxidative stress [20] (Figure 10B), and (iii) oxidative stress triggers the formation of reducible ATG7-containing higher-molecular-weight complexes (Figure 10C). Given that, (i) active autophagy requires transient thioester formation between LC3 and the catalytic thiols of ATG3 and ATG7 [18], (ii) oxidative stress can lead to disulfide bond formation between the catalytic thiols of ATG3 and ATG7, thereby preventing autophagosome maturation and blocking autophagy [18], and (iii) enhanced levels of po-H_2_O_2_ do not appear to stimulate autophagy (Figure 5). We also confirmed po-H_2_O_2_-induced complex formation between ATG3 and ATG7 (Figure 11). Note that we were unable to detect LC3B-bound ATG7 complexes. However, a likely explanation is that, under the conditions employed, the epitope recognized by the monoclonal anti-ATG7 antibody is masked.

### 3.6. OPTN Can Be Recruited to Peroxisomes and Trigger Pexophagy

To investigate the significance of our observation that OPTN can sporadically be observed on peroxisomes under basal growth conditions, we carried out additional immunofluorescence and live-cell imaging experiments. From these studies, it can be concluded that both endogenous as well as ectopically expressed OPTN-GFP can be recruited to peroxisomes (Figure 12). This was further evidenced by a plasma-membrane-targeted anti-GFP nanobody (GFP_Nb_-PM)-based translocation assay (Figure 13), in which a GFP-tagged protein of interest is relocated to the plasma membrane together with its (soluble or membrane-bound) interaction partner(s) [39]. Last but not least, overexpression of OPTN-GFP resulted in a significant decrease in peroxisome number (Figure 12), a phenomenon coinciding with pexophagy induction (Figure 14).

## 4. Discussion

In an analogy with mitochondria [47], pexophagy can be expected to play an important role in the clearance of peroxisomes under basal (e.g., for peroxisomal housekeeping), programmed (e.g., in different cell types during development), and stress-induced (e.g., for metabolic adjustments to external challenges) conditions. However, unlike in the mitophagy field, where multiple mitophagy activators are available [48], a big hurdle in the pexophagy field is the lack of small molecules or conditions that can be used as robust pexophagy inducers. Given that (i) mitophagy is intimately interwoven with mitochondrial oxidative stress [49], (ii) mitochondrial and peroxisomal abundance and activity are co-regulated at different levels [50], and (iii) peroxisomes produce significant amounts of H_2_O_2_ as part of their normal metabolism [51], it is reasonable to assume that disturbances in po-H_2_O_2_ metabolism may provoke signaling events that ultimately result in peroxisome degradation. Experimental observations supporting this hypothesis include the findings that (i) 1,10-phenanthroline-induced pexophagy in Chang liver cells coincides with elevated levels of po-H_2_O_2_, and that this process can be completely counteracted by N-acetyl-L-cysteine [52], (ii) interference with catalase activity stimulates pexophagy in serum-starved retinal pigment epithelial and HepG2 cells [53] as well as in the liver of mice subjected to prolonged fasting [54], and (iii) depletion of HSPA9, a predominantly mitochondrial heat-shock protein, in neuroblastoma cells induces pexophagy through an increase in po-H_2_O_2_ [55]. However, none of these studies provide direct evidence that elevated levels of po-H_2_O_2_ act as the primary trigger for pexophagy.

To find evidence in favor of or against the hypothesis that po-H_2_O_2_ can indeed act as a primary trigger of pexophagy, we generated DD-DAO Flp-In T-REx 293 and HeLa cell lines stably expressing po-mKeima and validated these in vitro models as a suitable tool for assessing pexophagy. Note that peroxisome-targeted variants of mKeima have already been used by others to study how alterations in the expression levels, subcellular localization, and/or activity of the ubiquitin-specific protease USP30 and HSPA9 affect peroxisome turnover [55,56,57,58]. During the validation process, we noticed high cell-to-cell variability in pexophagy, both under basal (Figure 1C) and starvation (Figure 2A) conditions. Given that such variability may complicate and even jeopardize a correct interpretation of the data, the levels of pexophagy were also routinely quantified using flow cytometry. Yet, despite extensive testing, we were not able to obtain any evidence that enhanced po-H_2_O_2_ emission in itself can trigger pexophagy, at least not in the cell lines studied. To reconcile the apparently paradoxical conclusions from our study and the studies mentioned above, it is important to highlight that pexophagy induction by inhibition of catalase activity was only observed during prolonged starvation [53,54]. In addition, it cannot be ruled out that the increased levels of po-H_2_O_2_ observed during 1,10-phenanthroline- or HSPA9-depletion-induced pexophagy [52,55] are an indirect consequence of other redox-related events that may impact pexophagy. Indeed, 1,10 phenanthroline is a high-affinity chelator for divalent metal ions such as Fe^2+^ and Zn^2+^, which are essential elements in many cellular processes, and HSPA9 is a multipotent chaperone regulating cellular processes ranging from viral infection to neurodegeneration [59]. In line with this, po-H_2_O_2_ is also unlikely to function as the primary pexophagy initiator in other conditions linking peroxisome degradation to increased peroxisomal oxidant levels. For example, in a study documenting that Hsc70/Stub1 promotes the removal of individual oxidatively stressed peroxisomes, the organelles were stressed by green light illumination after expression of a peroxisome-targeted variant of KillerRed, a condition predominantly generating superoxide anion radicals [60]. In addition, other studies showing that (i) pejvakin, a peroxisome-associated protein from the gasdermin family, triggers the removal of noise-induced oxidative-stress-damaged peroxisomes in auditory hair cells [61], and (ii) peroxisomal oxidative stress activates ATM and represses mTORC1 to induce autophagy and pexophagy [62], used external H_2_O_2_ to produce oxidative stress. However, it is essential to point out here that findings obtained with external H_2_O_2_, even at concentrations as low as 10 μM, cannot simply be extrapolated to conditions in which this oxidant is produced inside peroxisomes [63], and that such treatment can also promote autophagic cell death [64]. Finally, multiple studies have demonstrated that long-term inhibition of catalase activity even leads to an increase in peroxisome number [65,66,67], likely through an oxidative-stress-related decrease in autophagy. 

Currently, it is well known that all stages of autophagy are redox-regulated [11]. The results presented here provide compelling evidence that po-H_2_O_2_ has the potential to modulate autophagy at the levels of cargo recognition (Figure 8 and Figure 9) and phagophore expansion (Figure 10 and Figure 11). Although the (patho)physiological relevance of these findings remains to be clarified, it is known that (i) disulfide-linked oligomerization of SQSTM1, CACO2, and OPTN enables these SARs to bind to many ubiquitin tags on a single structure and to increase their avidity for LC3B clusters on the autophagosomal surface, thereby driving the bending of the isolation membrane around the cargo [22,68,69], (ii) oxidation of the redox-sensitive Cys residues in the LC3 processing enzyme ATG4B inhibits its activity, thereby leading to improved stability of LC3-PE and increased autophagosome formation [19], and (iii) oxidation of the catalytic thiols of ATG3 and ATG7 inhibits their activity in LC3–PE conjugation, thereby impairing autophagic flux [18]. In this context, our observation that excessive levels of po-H_2_O_2_ cause late-stage impairment in autophagy (Figure 7), a finding in line with the recent observation that catalase-deficient mice age faster through lysosomal dysfunction and defects in autophagosome-lysosome fusion [70], may not come as a complete surprise.

This study also revealed two unexpected but interesting side observations: (i) DAO-mediated production of po-H_2_O_2_ induced acidification of the peroxisome lumen (Figure 4C,D), and (ii) OPTN can be recruited to peroxisomes and trigger pexophagy (Figure 12, Figure 13 and Figure 14). Regarding the first observation, it must be mentioned that DAO is an enzyme that oxidizes the NH_2_-group of D-amino acids, thereby producing the corresponding α-keto acid, hydrogen peroxide, and ammonia. Given that enhanced levels of ammonia can be expected to result in the alkalinization of the peroxisomal matrix, the decrease in intraperoxisomal pH is most likely a (not yet explainable) H_2_O_2_-mediated phenomenon. Regarding the second observation, OPTN is a multifunctional autophagy receptor that contributes to mitophagy, aggrephagy, and xenophagy through ubiquitin signaling [71]. Our study showed that this receptor can also act as a pexophagy inducer, an important novel finding that needs further investigation. Pertinent follow-up research objectives, which fell outside the scope of this study, include determining the (i) conditions and regulatory mechanisms for recruitment of OPTN to the peroxisomal membrane, (ii) (ubiquitinated) binding partners of OPTN at the peroxisomal membrane, (iii) the role of OPTN in peroxisome function and clearance, and (iv) peroxisome abundance and metabolism in OPTN-related diseases.

## 5. Conclusions

To investigate the direct relationship between po-H_2_O_2_ emission and pexophagy, a topic not examined thus far, we established HEK-293 and HeLa pexophagy reporter cell lines in which the production of po-H_2_O_2_ can be controlled in a dose- and time-dependent fashion. Our results demonstrate that neither mild, moderate, nor high levels of po-H_2_O_2_ act as a primary trigger for pexophagy. However, po-H_2_O_2_ has the potential to oxidatively modify redox-sensitive SARs (e.g., SQSTM1, CACO2, and OPTN) and core autophagy proteins (e.g., ATG3, ATG4B, and ATG7), and enhanced levels of po-H_2_O_2_ potently inhibit late-stage autophagy. Finally, this study provides compelling evidence that OPTN can act as a pexophagy receptor, a novel finding that opens new avenues for exploring and exploiting the importance of peroxisome clearance in normal physiology and human disease.

## Figures and Tables

**Figure 1 antioxidants-12-00613-f001:**
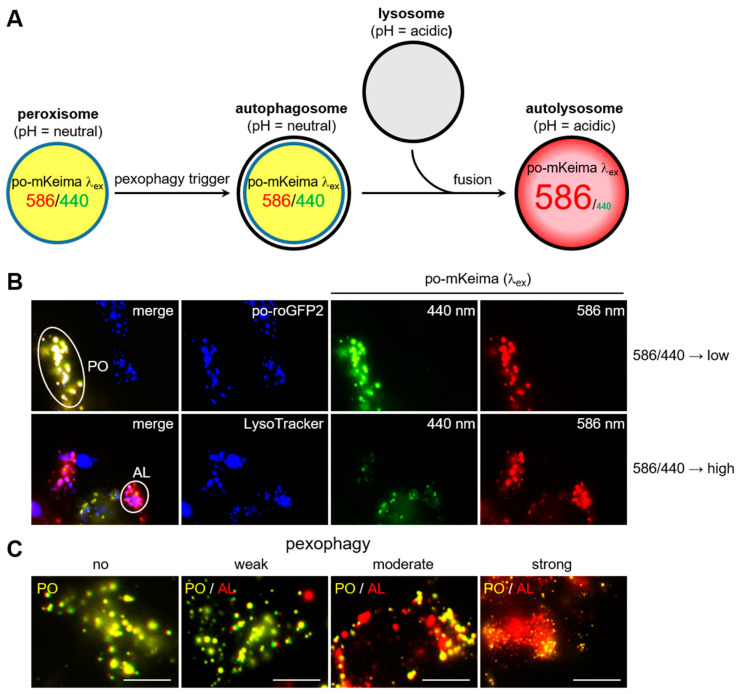
Pexophagy detection in living cells using po-mKeima. (**A**) Schematic outline. (**B**) Colocalization analysis between the peroxisomal marker po-roGFP2 (false color: blue) or the acidotropic fluorescent probe LysoTracker (false color: blue) and po-mKeima excited at 440 nm (false color: green) or 586 nm (false color: red). The yellowish (=low 586/440 excitation peak ratio) and reddish (=high 586/440 excitation peak ratio) dots represent peroxisomes (PO) and autolysosomes (AL), respectively. (**C**) Examples of DD-DAO/po-mKeima Flp-In T-REx 293 cells displaying no, weak, moderate, or excessively high levels of pexophagy. Scale bar, 10 µm.

**Figure 2 antioxidants-12-00613-f002:**
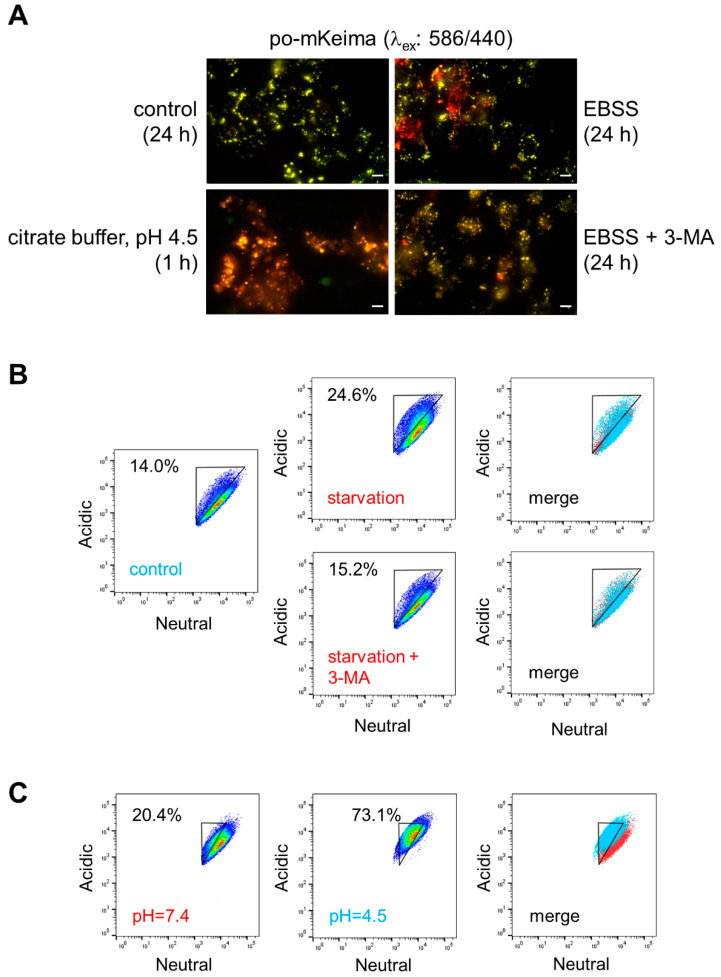
In situ detection of starvation-induced pexophagy. DD-DAO Flp-In T-REx 293 cells stably expressing po-mKeima were precultured for two days in rMEMα. At time zero, the medium was replaced with pre-warmed rMEMα (control), Earle’s balanced salt solution (EBSS) supplemented or not with 5 mM 3-methyladenine (3-MA), or a citrate buffer (pH 4.5). After the indicated times, the cells were processed for fluorescence microscopy or FACS analysis. (**A**) Representative images of the po-mKeima staining patterns in each group. Images were captured as described in the Section 2. Scale bar, 10 µm. (**B,C**) Representative flow cytometry plots of each group. Note that a decrease or increase in the fluorescence intensities of the neutral and acidic channels can be used to gate and quantify the percentage of cells undergoing pexophagy (starvation) and acidification of the peroxisome lumen (citrate buffer, pH 4.5), respectively. The different colors represent the cell density at a given position.

**Figure 3 antioxidants-12-00613-f003:**
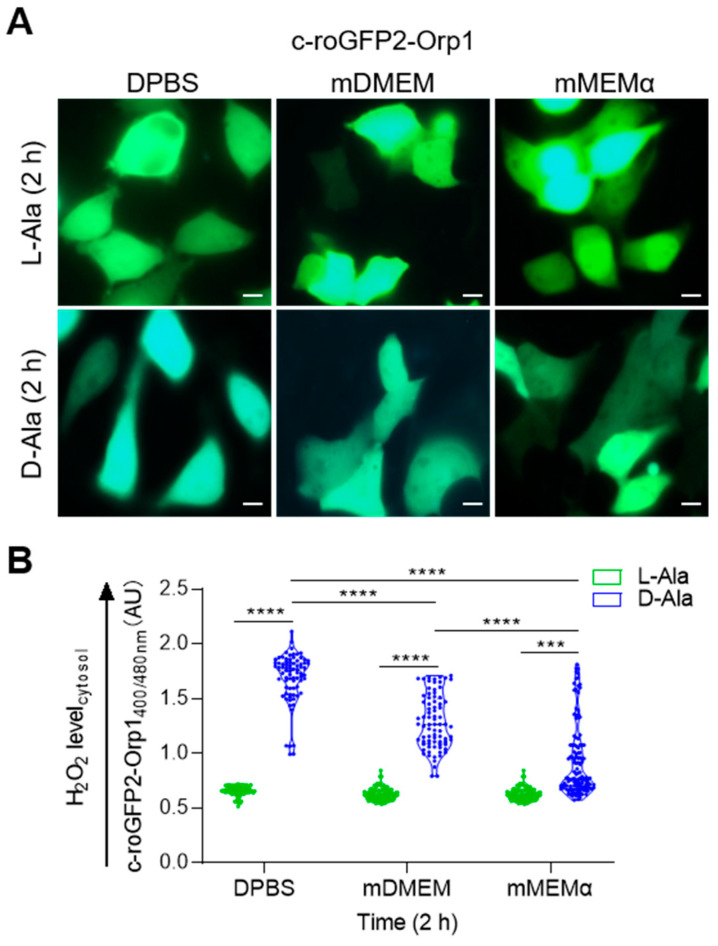
Effect of assay medium on peroxisomal H_2_O_2_ release. DD-DAO Flp-In T-REx 293 cells were cultured for 3 days in rMEMα containing 1 µg/mL DOX and 500 nM Shield1 (for details, see Section 2). Next, the cells were transfected with a plasmid encoding c-roGFP2-Orp1, a highly responsive cytosolic H_2_O_2_ sensor, and chased for one day in rMEMα without DOX/Shield1. Thereafter, the cells were incubated in different assay media (DPBS, mDMEM, or mMEMα; for details, see Section 2) and, after the addition of L- or D-Ala, the cytosolic H_2_O_2_ levels were monitored over time (the results obtained after 4, 8, and 24 h can be found in supplemental Appendix A). (**A**) Representative fluorescence overlay images of c-roGFP2-Orp1 on excitation at 400 (false color: blue) and 480 (false color: green) nm. Scale bar, 10 µm. (**B**) Response ratios of c-roGFP2-Orp1 after 2 h treatment. The data obtained for the D-Ala conditions were statistically compared as indicated in the graph (***, *p* < 0.001; ****, *p* < 0.0001).

**Figure 4 antioxidants-12-00613-f004:**
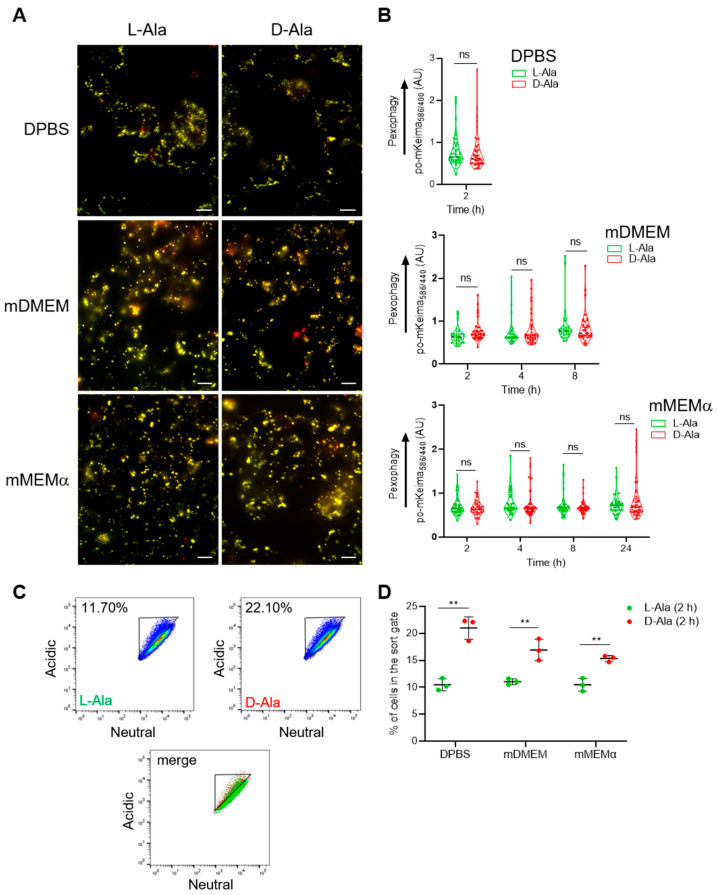
Peroxisome-derived H_2_O_2_ does not trigger pexophagy in po-mKeima/DD-DAO Flp-In T-REx 293 cells. DD-DAO Flp-In T-REx 293 cells stably expressing po-mKeima were cultured as described in the legend to Figure 3. The cells were subsequently transferred to different assay media (DPBS, mDMEM, or mMEMα; for details, see Section 2) supplemented with 10 mM L- or D-Ala. (**A**) Representative fluorescence overlay images of po-mKeima on excitation at 440 (false color: green) and 586 (false color: red) nm. Scale bar, 10 µm. (**B**) The po-mKeima 586/440 nm excitation fluorescence ratios. The data represent the values of the independent biological replicates. The corresponding D- and L-Ala data were statistically compared, but no significant differences were found. (**C**) Examples of flow cytometry plots for cells transferred to DPBS for 2 h. The different colors represent the cell density at a given position. (**D**) Quantification of the percentage of cells in the gated area. Data are shown as the mean ± SD (n = 3). The corresponding D- and L-Ala data were statistically compared (ns, non-significant; **, *p* < 0.01).

**Figure 5 antioxidants-12-00613-f005:**
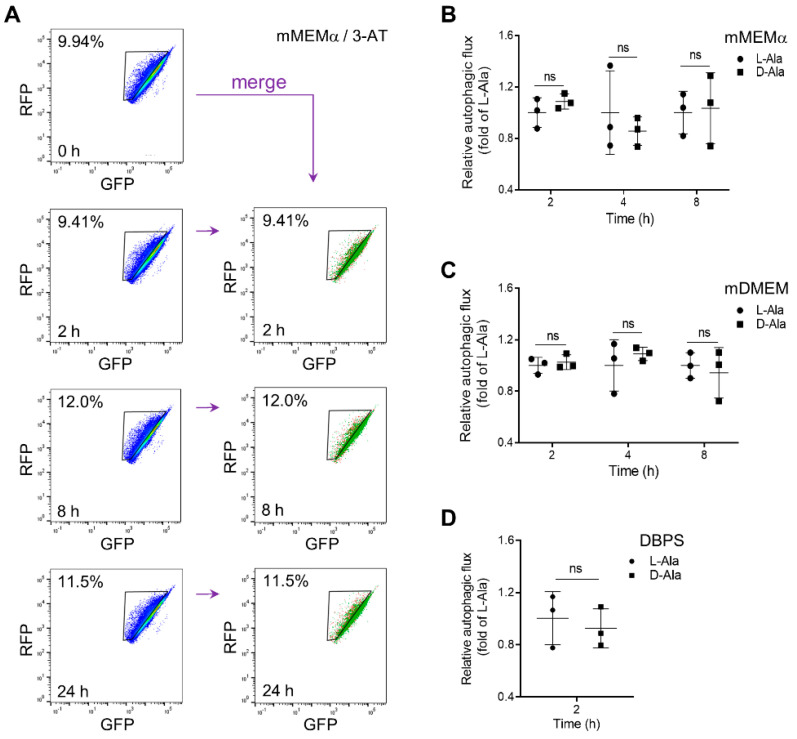
Enhanced levels of po-H_2_O_2_ do not induce autophagy in mRFP-GFP-LC3B DD-DAO Flp-In T-REx 293 cells. DD-DAO Flp-In T-REx 293 cells stably expressing RFP-GFP-LC3 were cultured as described in the legend to Figure 3. Next, the cells were assayed in (**A**) mMEMα supplemented with 10 mM 3-amino-1,2,4-triazole (3-AT), (**B**) mMEMα supplemented with 10 mM L- or D-Ala, (**C**) mDMEM supplemented with 10 mM L- or D-Ala, or (**D**) DPBS containing 10 mM L- or D-Ala (for medium details, see Section 2). At the indicated time points, the cells were harvested and processed for FACS analysis. To measure the percentage of cells undergoing autophagy, the RFP-GFP-LC3 single-cell populations were gated for a decrease in GFP expression. The results are shown as flow cytometry plots (n = 1; the different colors represent the cell density at a given position.) or interleaved scatter plots (n = 3; data represent the means ± SD). The corresponding D- and L-Ala data were statistically compared (ns, non-significant).

**Figure 6 antioxidants-12-00613-f006:**
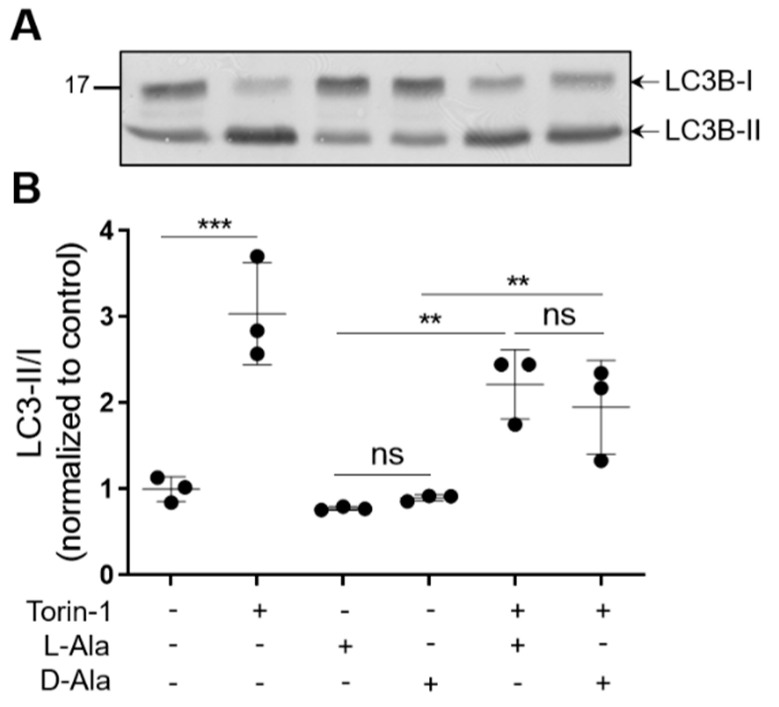
Moderate levels of po-H_2_O_2_ do not affect the processing of LC3B-I to LC3B-II. DD-DAO Flp-In T-REx 293 cells were cultured as described in the legend to Figure 3 and pre-incubated or not pre-incubated with 1 µM Torin-1 for 8 h. Thereafter, the cells were transferred to mMEMα (for details, see Section 2) supplemented with 10 mM L-Ala, 10 mM D-Ala, and/or 1 µM Torin-1. After 2 h, the cells were processed for immunoblotting with an antibody specific for LC3B. (**A**) Representative immunoblot. (**B**) Quantification of the LC3B-II/LC3B-I ratios. The data represent the means ± SD of three independent experiments. ***, *p* < 0.005; **, *p* < 0.01; ns, non-significant.

**Figure 7 antioxidants-12-00613-f007:**
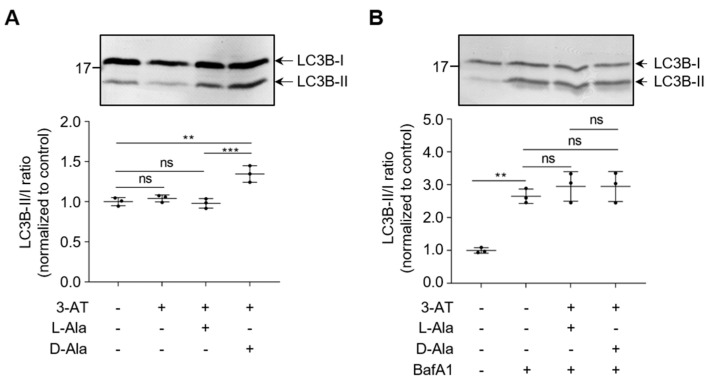
Severe po-H_2_O_2_ insults cause late-stage impairment of autophagy. DD-DAO Flp-In T-REx 293 cells were cultured as described in the legend to Figure 3. The cells, pre-incubated (**B**) or not pre-incubated (**A**) with 200 nM BafA1 for 8 h before the start of the experiment, were treated as indicated for 60 min and processed for immunoblotting with an antibody specific to LC3B. Representative immunoblots (upper panels) and quantifications (lower panels) are shown. Data represent means ± SD (n = 3 independent biological replicates). **, *p* < 0.01; ***, *p* < 0.005; ns, non-significant.

**Figure 8 antioxidants-12-00613-f008:**
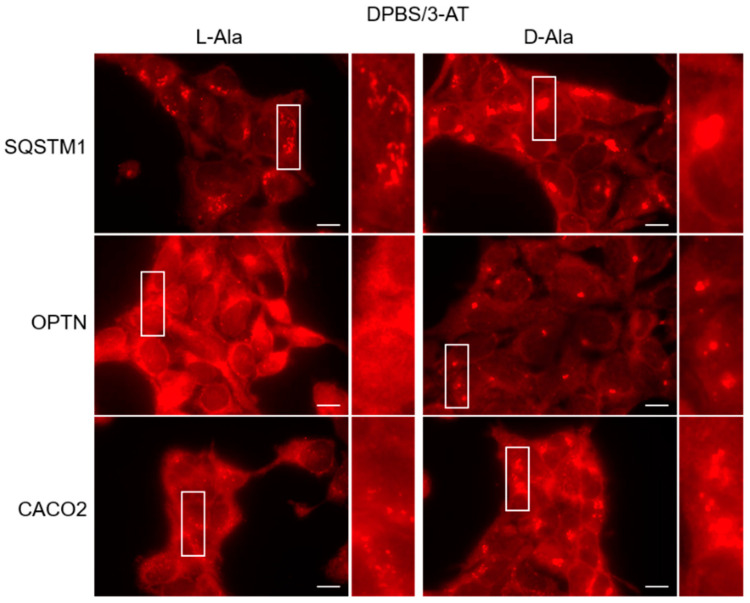
The specific autophagy receptors SQSTM1, OPTN, and CACO2 aggregate in response to peroxisome-derived H_2_O_2_ production. DD-DAO Flp-In T-REx 293 cells were cultured as described in the legend to Figure 3. The cells were subsequently transferred to DBPS containing 10 mM 3-amino-1,2,4-triazole (3-AT) and 10 mM L- or D-Ala. After 60 min, the cells were fixed and processed for immunofluorescence microscopy using rabbit antisera against SQSTM1, OPTN, or CACO2 in combination with a goat anti-rabbit secondary antibody conjugated to Texas Red. The panels shown on the right of each image are magnified views of the boxed areas. Scale bars, 10 μm.

**Figure 9 antioxidants-12-00613-f009:**
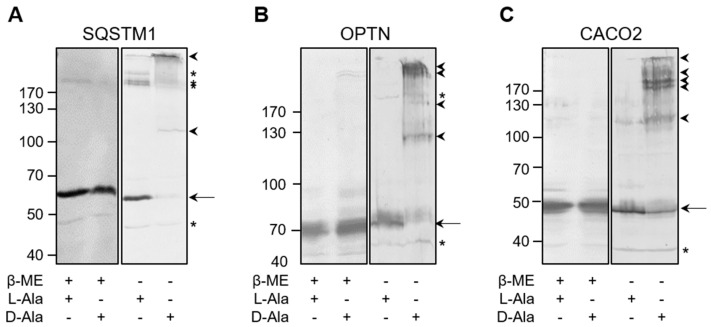
The specific autophagy receptors SQSTM1, OPTN, and CACO2 are prone to oxidation by peroxisome-derived H_2_O_2_. DD-DAO Flp-In T-REx 293 cells were cultured as described in the legend to Figure 3. Next, the cells were incubated in DBPS containing 10 mM 3-amino-1,2,4-triazole and 10 mM L- or D-Ala. After 60 min, the free thiol groups were blocked with N-ethylmaleimide. Thereafter, the cells were processed for SDS-PAGE under non-reducing (-β-ME) or reducing (+β-ME) conditions and subsequently subjected to immunoblot analysis with antibodies specific for (**A**) SQSTM1, (**B**) OPTN, or (**C**) CACO2. The migration points of relevant molecular mass markers (expressed in kDa) are shown on the left. The arrows and arrowheads mark the non-modified and oxidatively modified proteins, respectively. The asterisks mark bands of unknown nature.

**Figure 10 antioxidants-12-00613-f010:**
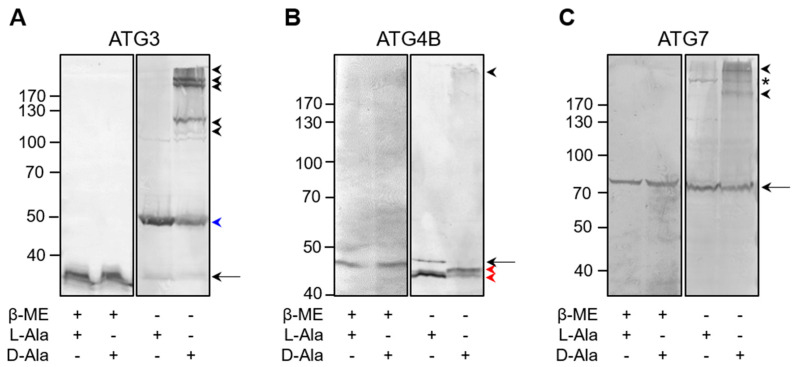
The core autophagy machinery proteins ATG3, ATG4B, and ATG7 are prone to oxidation by peroxisome-derived H_2_O_2_. DD-DAO Flp-In T-REx 293 cells were cultured as described in the legend to Figure 3. Next, the cells were incubated in DBPS containing 10 mM 3-amino-1,2,4-triazole and 10 mM L- or D-Ala. After 60 min, the free thiol groups were blocked with N-ethylmaleimide. Thereafter, the cells were processed for SDS-PAGE under non-reducing (-β-ME) and reducing (+β-ME) conditions and subsequently subjected to immunoblot analysis with antibodies specific for (**A**) ATG3, (**B**) ATG4B, or (**C**) ATG7. The migration points of relevant molecular mass markers (expressed in kDa) are shown on the left. The arrows and arrowheads mark the non-modified and oxidatively modified proteins, respectively. The blue and red arrowheads indicate, respectively, the disulfide-bonded ATG3-LC3 complex [18] and intramolecular disulfide-bonded forms of ATG4B [20]. The asterisk marks a band of unknown nature.

**Figure 11 antioxidants-12-00613-f011:**
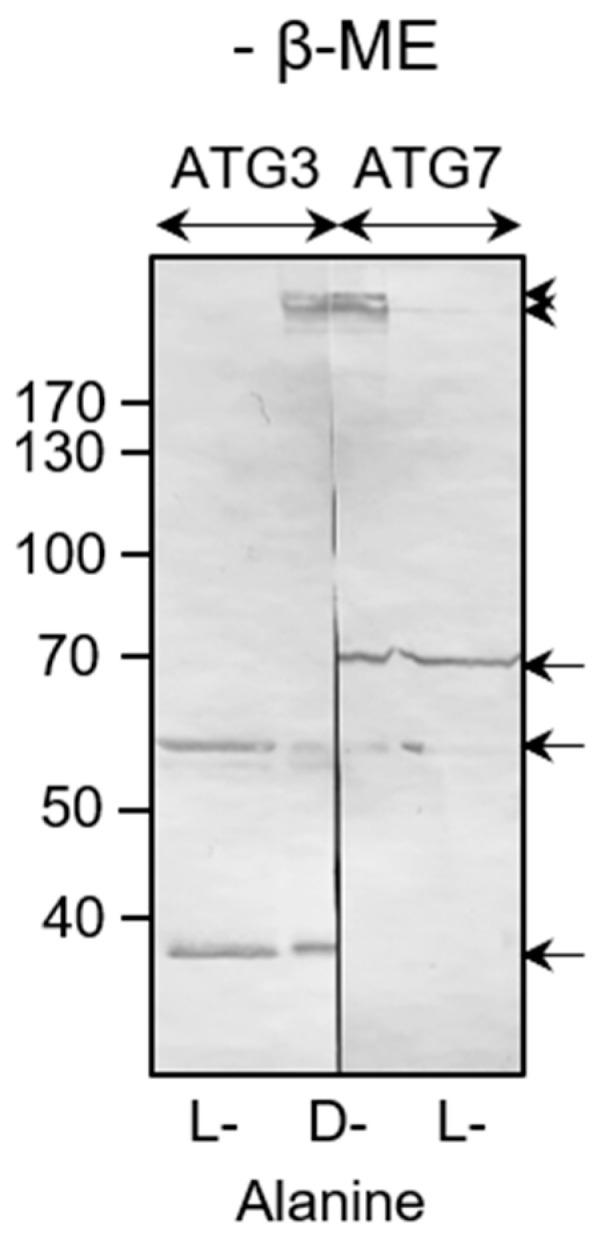
Peroxisome-derived H_2_O_2_ triggers complex formation between ATG3 and ATG7. DD-DAO Flp-In T-REx 293 cells were cultured as described in the legend to Figure 3. Next, the cells were incubated in DBPS containing 10 mM 3-amino-1,2,4-triazole and 10 mM L- or D-Ala. After 60 min, the free thiol groups were blocked with N-ethylmaleimide. Thereafter, the cells were processed for non-reducing SDS-PAGE and subsequently subjected to immunoblot analysis with antibodies specific for ATG3 or ATG7. Note that before probing the blots, the D-Ala lane was cut in two to unambiguously identify ATG3/ATG7-containing complexes. The migration points of relevant molecular mass markers (expressed in kDa) are shown on the left. The arrows and arrowheads mark non-oxidatively and oxidatively modified immunoreactive protein bands, respectively.

**Figure 12 antioxidants-12-00613-f012:**
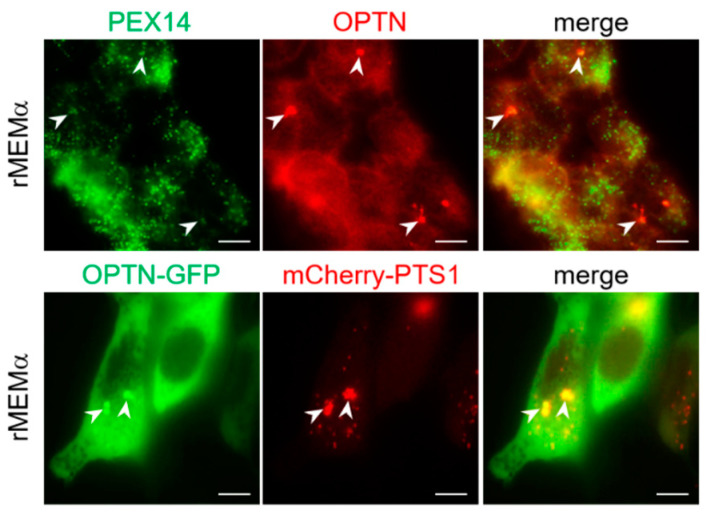
Endogenous and GFP-tagged OPTN can be recruited to peroxisomes under basal growth conditions. DD-DAO Flp-In T-REx 293 cells were co-transfected (**lower panels**) or not co-transfected (**upper panels**) with plasmids encoding OPTN-GFP and mCherry-PTS1. After culturing the cells for two days in rMEMα, they were processed for live-cell imaging (**lower panels**) or fixed and processed for immunofluorescence microscopy using a mouse antiserum against PEX14 and a rabbit antiserum against OPTN, in combination with goat anti-mouse and anti-rabbit secondary antibodies conjugated to Alexa Fluor 488 and Texas Red, respectively (**upper panels**). The white arrowheads point to sites of colocalization between OPTN(-GFP) and the peroxisomal marker protein PEX14 or mCherry-PTS1. Scale bar, 10 µm.

**Figure 13 antioxidants-12-00613-f013:**
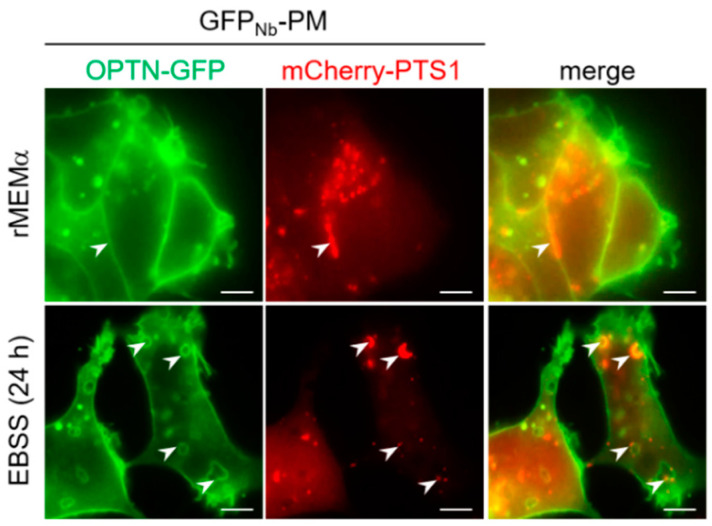
Co-expression of OPTN-GFP and a plasma-membrane-targeted anti-GFP nanobody recruits peroxisomes to the plasma membrane. DD-DAO Flp-In T-REx 293 cells were transiently co-transfected with plasmids encoding a plasma membrane targeting motif-tagged anti-EGFP nanobody (GFP_Nb_-PM), OPTN-GFP, and mCherry-PTS1, then cultured in rMEMα. The next day, the medium was replaced with fresh medium or EBSS and, 24 h later, the cells were imaged using fluorescence microscopy. Representative images are shown. The white arrowheads indicate positions where peroxisomes are associated with the plasma membrane. Scale bar, 10 µm.

**Figure 14 antioxidants-12-00613-f014:**
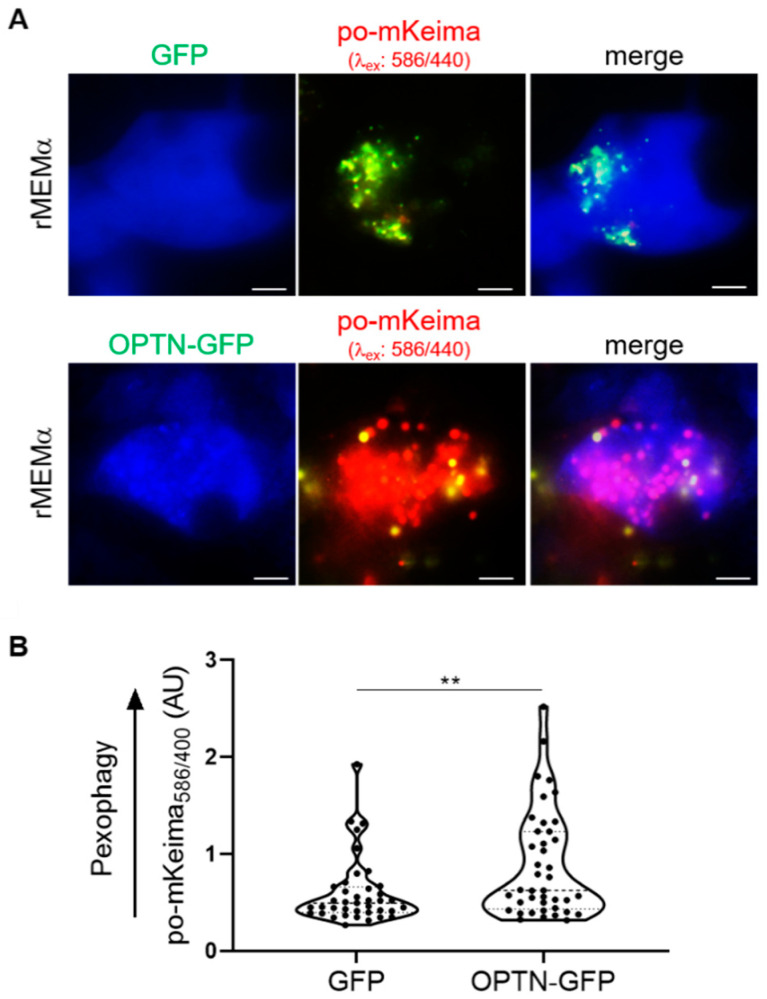
Expression of OPTN-GFP triggers pexophagy. DD-DAO Flp-In T-REx 293 cells expressing po-mKeima were first transfected with a plasmid encoding GFP or OPTN-GFP and subsequently cultured in rMEMα. Two days later, the cells were processed for live-cell imaging. (**A**) Representative fluorescence images of (OPTN-)GFP (false color: blue) and po-mKeima on excitation at 440 (false color: green) and 586 (false color: red) nm. Scale bar, 10 µm. (**B**) Random fields of view were selected in the bright field, and the 586/440 nm fluorescence excitation ratios of po-mKeima were measured and plotted. **, *p* < 0.01.

## Data Availability

Not applicable.

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
