# Peer review of "Enhanced Levels of Peroxisome-Derived H_2_O_2_ Do Not Induce Pexophagy but Impair Autophagic Flux in HEK-293 and HeLa Cells"

_antioxidants, 2023, doi:10.3390/antiox12030613_

Round 1

Reviewer 1 Report

In this manuscript, Fransen and colleagues investigated how peroxisome-derived H2O2 influences pexophagy and macroautophagy in mammalian cell cultures. A novel peroxisome-targeted mKeima reporter (po-mKeima) was generated to monitor pexophagy. The authors found that peroxisome-derived H2O2 did not alter pexophagy, but did block autophagic flux, potentially through the oxidation of autophagy receptors and key proteins in autophagosome formation. The authors also discovered that the autophagy receptor optineurin co-localizes with peroxisomes to regulate pexophagy. This study provides important insights into the role of peroxisome-derived H2O2 in the cellular autophagic process. The manuscript is well prepared and presented. The conclusions are sound and supported by a large amount of cellular and biochemical evidence. I only have a few minor concerns as listed below.

1. Figure 7, to support the conclusion that po-H2O2 inhibits autophagy at a later stage, the authors should run the proteins with or without BafA1 treatment on the same protein gel, to obtain a clear comparison. Based on the current results, there was still some autophagic activity in the severe po-H2O2 group (the LC3B-II/I ratio was 2.8 and 1.4 with or without BafA1, respectively), which is similar to the LC3B-II/I ratio of the control group (2.6 and 1.0 with or without BafA1).

The authors could also run WB to quantify the levels of autophagy adaptor proteins (SQSTM1, CACO2, and OPTN) under reducing conditions. The accumulation of these adaptor proteins can be another indicator of impaired autophagy. In fact, their fluorescent imaging results in Figure 8 show that both fluorescent intensities of SQSTM1 and CACO2 were induced by severe po-H2O2, suggesting that autophagic flux could be blocked. Interestingly, OPTN intensity was reduced.

2. The observations of OPTN recruitment to peroxisomes are very intriguing. I wonder if the authors have looked at whether the co-localization OPTN and Pex14 increases under pexophagy induction conditions, such as amino-acid starvation. Could OPTN knockdown block stress-induced pexophagy?

Author Response

We thank to the reviewer for his/her positive comments and appreciation of our work.

Figure 7, to support the conclusion that po-H2O2 inhibits autophagy at a later stage, the authors should run the proteins with or without BafA1 treatment on the same protein gel, to obtain a clear comparison. Based on the current results, there was still some autophagic activity in the severe po-H2O2 group (the LC3B-II/I ratio was 2.8 and 1.4 with or without BafA1, respectively), which is similar to the LC3B-II/I ratio of the control group (2.6 and 1.0 with or without BafA1).

As requested, we now include data from samples that were analyzed on the same protein gel (see Figure S9). As can be seen in this figure, the data essentially recapitulate the results shown in Figure 7. As such, the reviewer is correct that we cannot claim that high levels of po-H2O2 “block” autophagy at a later stage. However, from our data, it is clear that high levels of po-H2O2 do “impair” autophagy at a later stage. In this context, it is also important to realize that, while the D-Ala treatment only took 60 min, the cells were treated for 8 h with BafA1. As such, even a complete block by po-H2O2 would result in lower LC3B-II/I ratios in the “+3-AT/+D-Ala” condition compared to the “+BafA1/+3-AT/+D-Ala” condition. Nevertheless, to avoid our conclusions being overstated, we have replaced the term “late-stage block” by “late-state impairment” (see lines 324, 365, and 551 in the marked version of the manuscript).

The authors could also run WB to quantify the levels of autophagy adaptor proteins (SQSTM1, CACO2, and OPTN) under reducing conditions. The accumulation of these adaptor proteins can be another indicator of impaired autophagy. In fact, their fluorescent imaging results in Figure 8 show that both fluorescent intensities of SQSTM1 and CACO2 were induced by severe po-H2O2, suggesting that autophagic flux could be blocked. Interestingly, OPTN intensity was reduced.

Thank you for the suggestion. However, here it is important to note that (i) immunofluorescence images only provide semi-quantitative results, and (ii) the data shown in Figure 8 were obtained after a 1-h incubation, which is most likely not long enough to detect significantly enhanced autophagy adaptor protein levels. However, in deference to the reviewer, we also analyzed the levels of SQSTM1 in the samples that were used to prepare Figure S9. As can be seen in the “Figure for review only”, although the levels of SQSTM1 are upregulated in cells pretreated for 8 h with BafA1, no significant differences can be observed in the samples treated for 1 h with D-Ala. Given (i) the journal’s request to revise the manuscript within 5 days, (ii) the limited amounts of samples that we had left, and (iii) we do not expect to see any significantly increased autophagy adaptor protein levels within the 1-h time window under investigation, we hope the reviewer finds it acceptable that we didn’t analyze the CACO2 and OPTN levels. In addition, given the likely negative outcome of this experiment, we do not prefer to include this figure in the manuscript.

The observations of OPTN recruitment to peroxisomes are very intriguing. I wonder if the authors have looked at whether the co-localization OPTN and Pex14 increases under pexophagy induction conditions, such as amino-acid starvation. Could OPTN knockdown block stress-induced pexophagy?

These are very relevant and pertinent questions! However, as pointed out in the discussion of the manuscript (lines 564-570 in the marked version of the manuscript), we feel that the further characterization of OPTN-dependent pexophagy is outside the scope of this study, which focuses on how po-H2O2 affects general autophagy and pexophagy. We hope the reviewer can agree with this point of view.

Reviewer 2 Report

The study of Hong Li pure et al. investigated the role of peroxisome-derived H2O2 (po- H2O2) and its associated signaling in the general cellular autophagy and more specifically in the peroxisome autophagic process (pexophagy). To this end, the authors created an elegant reporter system based on the controlled expression of a mutant peroxisomal D-amino acid oxidase (DAO) ; and cell lines expressing a peroxisome-targeted variant of mKeima (po-mKeima), a red fluorescent protein, which has different emission peaks depending on the pH.

The rational of this study is of high significance to decipher particularly peroxisome-derived H2O2 role in cell autophagy. The authors take the precaution in using two different cells lines expressing the rescued DAO mutant and po-mKeima Flp-In T-Rex reporter to monitor both the peroxisomal H2O2 production and to fellow pexophagie process. Although the reported data are dense with cumulative cell treatments, the authors tried to keep the reader understanding the experiments progress and the deduced arguments.    

Few points to be clarified:

-       Line 249 : If you mean that the stressor is the starvation condition, please reformulate the sentence : “DD-DAO/po-mKeima Flp-In T-REx 293 cells were subjected to starvation in EBSS buffer, a stressor known to induce pexophagie”.

-       In Figure 3B, under D-Ala treatment, the level of H2O2 produced is similar in mDMEM or mMEMalpha medium, while in Figure 3A, the shift from the bright-green to the bright-blue is seen only in mDMEM. Such difference is not seen for HeLa cells in figure S3. How the authors explain such difference?

-       For the effect of assay medium on peroxisomal H2O2 release, in figure S2 the experiments were done at 4h, 8h and 24h and in figure S3 at 2h? Is this related to the nature of cell line? did the authors checked the cell viability?

-       Figure 7B, the control using both 3-AT and BafA1 is missing. Did the authors already try such condition?

Author Response

We thank the reviewer for his/her positive feedback and thoughtful comments, and we have attempted to address all issues raised.

Line 249 : If you mean that the stressor is the starvation condition, please reformulate the sentence: “DD-DAO/po-mKeima Flp-In T-REx 293 cells were subjected to starvation in EBSS buffer, a stressor known to induce pexophagy”.

The stressor is indeed the starvation condition (and not the buffer itself). To avoid any confusion, we omitted “in EBSS buffer” and the sentence reads now as follows: “To functionally validate po-mKeima, DD-DAO/po-mKeima Flp-In T-REx 293 cells were subjected to starvation, a stressor known to induce pexophagy [45].” (see line 249 in the marked version of the manuscript).

In Figure 3B, under D-Ala treatment, the level of H2O2 produced is similar in mDMEM or mMEMalpha medium, while in Figure 3A, the shift from the bright-green to the bright-blue is seen only in mDMEM. Such difference is not seen for HeLa cells in figure S3. How the authors explain such a difference?

The reviewer is correct when he/she is pointing out that the bright-green to bright-blue shift is hardly or not visible in the mMEMa panels shown in Figure 3A. However, as can be seen in panel 3B, the overall oxidation states of c-roGFP2-ORP1 are not the same in the mDMEM and mMEMa conditions. That is, while the oxidation state of this H2O2 sensor only increases weakly (£ 1.5 fold) in the majority (±65%) of the mMEMa-cultured cells (median increase per cell: 1.24-fold), it increases >1.5-fold in 95% of the mDMEM-cultured cells. In addition, as can be seen in the non-cropped images (see file with raw data), there are mMEMa-cultured cells displaying a bright-green to bright-blue shift. Importantly, to improve clarity for readers, we now also statistically compare the different D-Ala conditions to document that different assay media exhibit significant differences in cytosolic antioxidant capacity (see Figures 3B, S2A, S2B, and S3B).

With respect to Figure S3, we would like to point out that, on average, the oxidation state of c-roGFP2-roGFP2 is also >1.50-fold increased in 56% of the cells cultured for 2 h in D-Ala-containing mMEMa, thereby indicating that a similar D-Ala treatment results in slightly higher H2O2 levels within the cytosol of HeLa cells than within HEK-293 cells. In addition, it is important to keep in mind that pixel intensity comparisons are superior to visual image analyses, which only yield semi-quantitative information (thereby increasing the risk of not detecting small differences).

For the effect of assay medium on peroxisomal H2O2 release, in figure S2 the experiments were done at 4h, 8h and 24h and in figure S3 at 2h? Is this related to the nature of cell line? did the authors checked the cell viability?

The first series of experiments were carried out in HEK-293 cells and, given that we had no idea about what to expect, we collected data at 5 different time points (0, 2, 4, 8, and 24 h; see Figures 3 and S2). However, to extend our findings to HeLa cells, we only compared the extreme conditions (DPBS, 2 h; mMEMa, 2 and 8 h; see Figure S3). We did not check cell viability. However, as mentioned in the legend to Figure S2, only those conditions were quantified in which the cells show an overall healthy appearance (the underlying reason being that excessive levels of po-H2O2 cause cell detachment and damage).

Figure 7B, the control using both 3-AT and BafA1 is missing. Did the authors already try such a condition?

Given that we routinely include an “L-Ala negative control” for each condition in which we study the impact of po-H2O2 on a specific process (e.g., in this case, the processing of LC3-I to LC3-II) by D-Ala supplementation to DAO-expressing cells, we also added L-Ala to the “+3-AT/+BafA1” condition (see Figure 7B, lane 3). However, in deference to the reviewer, we now also include a “+3-AT/+BafA1” condition lacking L-Ala in the new experiment that was carried out on request of Reviewer 1 (see Figure S9, lane 2).

Reviewer 3 Report

The article entitled “Enhanced Levels of Peroxisome-Derived H2O2 Do Not Induce 2 Pexophagy but Impair Autophagic Flux in HEK-293 and HeLa cells and submitted to Antioxidant by Hongli Li et al. is of huge interest for the whole cell biology community and beyond. I did not find any interesting opportunity to modified this article, so it can be accepted as presented.

Author Response

We thank the reviewer for his/her appreciation and positive assessment of our manuscript.